# Renal Leukemic Infiltration Overlapping Acute Focal Bacterial Nephritis during Myelodysplastic Syndrome: An Autopsy Case Report

**DOI:** 10.3390/medicina58081060

**Published:** 2022-08-05

**Authors:** Keishu Murakami, Shinobu Tamura, Anna Maruyama, Tomomi Naitoh, Kan Teramoto, Yurina Mikasa, Masaoh Tanaka, Shinichi Murata, Seiya Kato

**Affiliations:** 1Department of Emergency and Critical Care Medicine, Wakayama Medical University, Wakayama 641-8509, Japan; 2Department of Neurology, Wakayama Medical University, Wakayama 641-8509, Japan; 3Department of Hematology/Oncology, Wakayama Medical University, Wakayama 641-8509, Japan; 4First Department of Medicine, Wakayama Medical University, Wakayama 641-8509, Japan; 5Department of Nephrology, Wakayama Medical University, Wakayama 641-8509, Japan; 6Department of Diagnostic Pathology, Wakayama Medical University, Wakayama 641-8509, Japan

**Keywords:** myelodysplastic syndrome, acute myeloid leukemia, acute focal bacterial nephritis, renal leukemic infiltration, wedge-shaped parenchyma

## Abstract

Renal leukemic infiltration is uncommon in myeloid neoplasms, including myelodysplastic syndromes (MDS). A 76-year-old male patient was admitted to our hospital with complaints of fever and dyspnea. He was diagnosed with MDS with multilineage dysplasia and acute focal bacterial nephritis (AFBN) based on clinical, laboratory, and radiological investigations. Antibiotic treatment temporarily improved his condition, but the radiological image of AFBN remained. His condition gradually deteriorated into multiple organ failure, and he unfortunately died on the 31st day of hospitalization. Autopsy findings revealed significantly increased p53-positive blasts in the bone marrow and renal parenchyma overlapping AFBN, suggesting leukemic transformation and renal infiltration. This case emphasizes the need to review the diagnosis when antibiotic treatment is ineffective in MDS patients with AFBN.

## 1. Introduction

Myelodysplastic syndromes (MDS) are a group of clonal hematopoietic stem cell diseases that are characterized by peripheral blood cytopenia, ineffective hematopoiesis, and the risk of transforming into acute myeloid leukemia (AML) [1]. The common causes of death in patients with MDS are AML transformation (46.6%), infection (27%), and bleeding (9.8%) [2]. Neutropenia occurs in nearly 60–80% of newly diagnosed patients with high-risk MDS [3]. Infectious complications in such patients interfere with the induction of the initial treatment, which can include hypomethylating agents, chemotherapy, or lenalidomide [3]. Therefore, infections associated with MDS, especially in patients with neutropenia, need to be appropriately diagnosed and treated based on clinical and laboratory findings.

Acute focal bacterial nephritis (AFBN) is a localized bacterial parenchyma infection that presents as an inflammatory mass without abscess formation [4]. A characteristic finding in computed tomography (CT) is inhomogeneous or wedge-shaped parenchyma with decreased enhancement [5]. However, this finding has also been reported as renal involvement in pediatric acute lymphoblastic leukemia (ALL) [6,7].

Renal leukemic infiltration is uncommon in myeloid neoplasms, including MDS. Herein, we present the case of an elderly patient with MDS treated with AFBN at initial admission who was finally diagnosed with renal leukemic infiltration and AML transformation by autopsy.

## 2. Case Report

A 76-year-old male patient was admitted to our hospital with a 1-day history of fever and dyspnea. He had a previous medical history of lung cancer, gastric cancer, and benign prostatic hyperplasia. On admission, his body temperature was 39.1 °C, blood pressure was 112/62 mmHg, pulse rate was 140 beats/min, and respiratory rate was 24 breaths/min. General physical examination showed pallor of the palpebral conjunctiva and right costovertebral angle tenderness.

Laboratory test results upon admission are summarized in Table 1. Complete blood cell count analysis revealed macrocytic anemia (hemoglobin of 5.4 g/dL) and thrombocytopenia (platelet count of 24 × 10^9^/L) with a normal range of white blood cells (6.6 × 10^9^/L). A peripheral blood smear examination showed 2.0% blasts, 1.0% metamyelocytes, and 6/100 erythroblasts. Biochemical analysis revealed mild renal impairment (serum creatinine level of 1.52 mg/dL), and high levels of lactate dehydrogenase (LDH; 768 IU/L) and C-reactive protein (CRP; 13.97 mg/dL). The D-dimer level was also elevated (80.04 μg/dL). Urine microscopy detected white blood cells and Gram-negative rods. Two peripheral blood cultures and a urinary culture were taken. Three days after admission, *Escherichia coli,* which is sensitive to all antibiotics, was isolated in the urinary culture. A contrast-enhanced CT showed a wedge-shaped lesion with decreased contrast enhancement in the right kidney (Figure 1A). The other infection focus was not observed in the CT findings.

Bone marrow aspiration at admission showed normocellular marrow (nuclear cell count of 188,000/µL, myeloid/erythroid ratio of 7.4, and megakaryocyte level of 45/µL). Morphological examination revealed a marked trilineage dysplasia and a low blast percentage (2.6%). These findings met the diagnostic criteria for MDS with multilineage dysplasia (MLD) according to the 2016 World Health Organization Classification [1]. Further, G-banding staining of metaphase chromosomes of bone marrow cells from our MDS-MLD patient showed complex karyotypes with more than three abnormalities, as follows: 47,XY,der(5;19)(p10;q10),+8,der(8;15)(q10;q10),+19,+mar1[1]/46,idem,−18[13]/46,XY [2]. The risk group was classified as very high (5 points) by the Revised International Prognostic Scoring System (IPSS-R) [1]. The first diagnosis was AFBN in a patient with very-high-risk MDS based on the clinical, laboratory, and radiological information.

He was treated with tazobactam/piperacillin (TAZ/PIPC) for AFBN and transfusions for MDS. Levofloxacin (LVFX) was administered instead of TAZ/PIPC on the ninth day of hospitalization because of persistent fever and marked CRP elevation. Afterward, his inflammatory condition temporally improved. On the 14th day, he had a fever up to 38.0 °C and elevated CRP again. Moreover, *Enterococcus faecium* was detected in his urine culture. Combination therapy with LVFX and vancomycin was selected, but the high fever continued. A contrast-enhanced CT image on the 15th day showed that the wedge-shaped lesion in the right kidney remained despite antibiotic administration (Figure 1B). On the 23rd day, voriconazole was started and LVFX was switched to meropenem. However, the patient’s condition deteriorated into multiple organ failure, and he finally died on the 31st hospitalization day. The peripheral blood leukemic cells did not exceed 10% during the hospitalization period. A pathological autopsy was performed shortly after death with written informed consent from the patient’s family. Figure 2 summarizes the patient’s clinical course.

The autopsy revealed systemic organ congestion, bilateral pleural cavity effusion, and retroperitoneal hemorrhage. The macroscopic examination of the right kidney did not show an abscess. The microscopic examination identified that sternal and vertebral bone marrow was filled with p53-positive blasts (Figure 3A–C), consistent with AML transformation. Consistent with the CT finding of wedge-shaped lesions, small nodules with central necrosis and invasion of many p53-positive blasts were found in the right kidney (Figure 3D–F). Furthermore, a few p53-positive blasts infiltrated into the aorta, liver, spleen, and skin. Therefore, our patient was finally diagnosed with leukemic infiltration of multiple organs, especially the right kidney.

## 3. Discussion

Renal leukemic infiltration has been reported in ALL and low-grade non-Hodgkin’s lymphoma [6,7,8,9]. Sabui et al., reported renal leukemic infiltration in 7–42% of ALL cases. These findings support the assumption that localized infiltration is relatively common in patients with ALL [10]. Meanwhile, renal leukemic infiltration is uncommon in patients with AML [11]. One report described renal leukemic infiltration in an elderly patient with MDS-transformed AML [12]. Here, we present renal leukemic infiltration overlapping AFBN without marked leukocytosis during the course of MDS. In such cases, making a diagnosis of renal leukemic infiltration is challenging when AML transformation in patients with MDS has not been diagnosed.

Most patients with ALL have bilateral renal infiltration, causing nephromegaly at diagnosis [6]. Meanwhile, no characteristic imaging findings of renal involvement in AML have been reported; some cases have shown renal enlargement on CT images [11], while one case did not show any imaging findings [12]. It has been reported that a 2-year-old child with ALL was initially diagnosed as having AFBN based on the CT findings, which showed wedged-shaped, low-attenuation masses in the bilateral kidneys [7]; however, to the best of our knowledge, there have been no reports of renal myeloblastic infiltration mimicking AFBN. Sieger et al., proposed that it is necessary to conduct an invasive procedure, such as a percutaneous puncture or surgical exploration, in patients with AFBN refractory to antibiotic therapy to rule out renal malignancies [5]. In our case, a contrast-enhanced CT image on the 15th hospitalization day revealed the remaining wedge-shaped lesion in the right kidney regardless of antibiotic administration, suggesting that renal leukemic infiltration occurred following the transformation of MDS to AML. At that point, such a patient should be considered for percutaneous renal puncture to review the AFBN diagnosis.

Most patients with MDS are at risk of dying from complications that are directly related to cytopenia or AML progression [13]. Among them, high serum LDH (>240 IU/L) has been associated with decreased overall survival (OS) and increased AML transformation [14]. The IPSS-R provides a useful prognostic score for clinical outcomes of untreated patients with MDS [15]. Furthermore, an elevated LDH level is a significant feature for predicting OS but not AML transformation [15]. Our patient showed transiently increased serum LDH on the 7th and 13th hospitalization days, which first suggested AFBN exacerbation. However, the serum LDH level was maintained at >700 U/L after antibiotic treatment was initiated. During hospitalization, this patient with high-risk MDS might have developed rapid AML progression. These findings support the idea that marked serum LDH elevation is highly suggestive of AML transformation. Therefore, although the blast ratio in the peripheral blood of our patient did not exceed 10%, it was necessary to repeat bone marrow aspiration during the hospitalization.

Azacitidine, a hypomethylating agent, is currently used as the initial treatment for patients with high-risk MDS who are ineligible for intensive chemotherapy and allogeneic hematopoietic stem cell transplantation (HSCT) [1]. After the AFBN was cured, our elderly patient was considered for azacitidine treatment. However, we missed an opportunity fotreatment intervention, because the MDS rapidly progressed to AML during this hospitalization. Recently, treatment with venetoclax, a selective BCL-2 inhibitor, plus azacitidine, has been reported to improve clinical outcomes in patients with newly diagnosed AML who are ineligible for intensive chemotherapy and HSCT [16]. Meanwhile, grade 3/4 febrile neutropenia occurred in 30% of patients receiving azacitidine plus venetoclax in a clinical trial [16]. With an accurate AML diagnosis during this hospitalization, our elderly patient might have been a good candidate for venetoclax plus azacitidine under empirical antibiotic treatment.

In summary, we report the first case of renal leukemic infiltration overlapping AFBN during MDS. Autopsy findings revealed an extremely rapid AML transformation in our patient with MDS-MLD. Leukemic involvement should be suspected in patients with MDS with remaining renal hypoperfusion and persistently elevated serum LDH levels despite the appropriate antibiotic treatment.

## 4. Conclusions

Here, we describe an autopsy case report of renal leukemic infiltration overlapping AFBN during MDS. On admission, the patient was diagnosed with AFBN and MDS-MLD; however, autopsy findings revealed significantly increased p53-positive blasts in bone marrow and renal parenchyma, suggesting leukemic transformation and renal infiltration. This case emphasizes that clinicians should review the diagnosis of MDS when antibiotic treatment is ineffective in patients with AFBN.

## Figures and Tables

**Figure 1 medicina-58-01060-f001:**
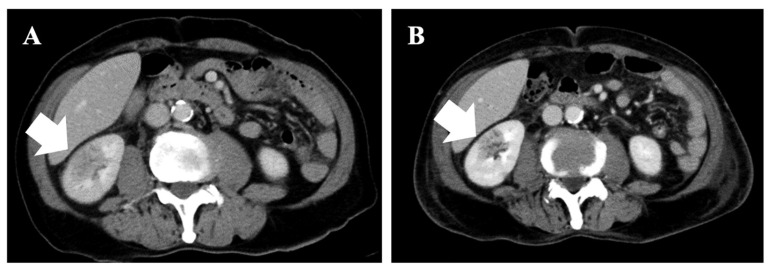
Images of contrast-enhanced abdominal computed tomography on (**A**) first and (**B**) 15th day of hospitalization. These images show wedge-shaped lesions with decreased contrast enhancement in right kidney (white arrows).

**Figure 2 medicina-58-01060-f002:**
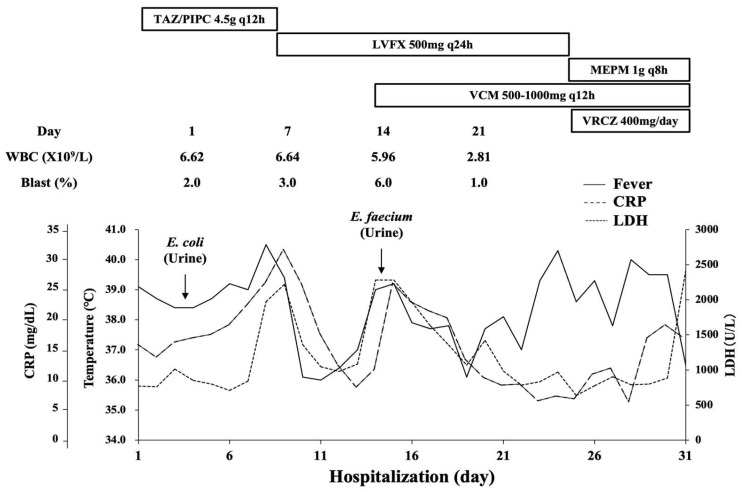
Patient’s clinical course after admission to our hospital. CRP, C-reactive protein; *E. coli*, *Escherichia coli*; *E. faecium*, *Enterococcus faecium*; LDH, lactate dehydrogenase; LVFX, levofloxacin; MEPM, meropenem; TAZ/PIPC, tazobactam/piperacillin; VCM, vancomycin; VRCZ, voriconazole; WBC, white blood cell.

**Figure 3 medicina-58-01060-f003:**
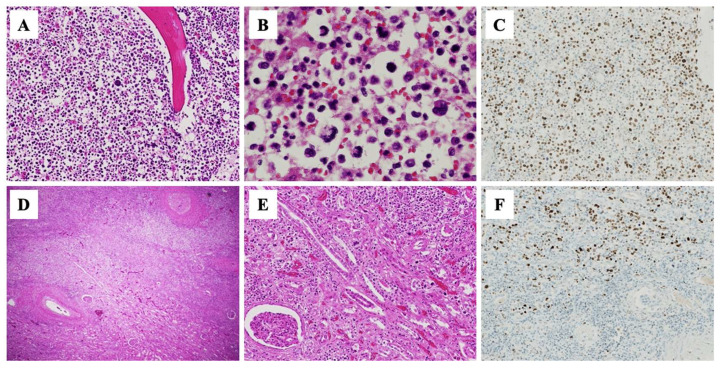
Histological findings of bone marrow and kidney on autopsy. Hematoxylin and eosin (H-E) staining of sternum shows increased blasts ((**A**) ×100; (**B**), ×400) consistent with leukemic transformation. Immunohistochemical staining was positive for p53 ((**C**), ×100). Small nodules with central necrosis ((**D**), ×20) and blast invasion in right kidney ((**E**), ×100) on H-E staining. p53 protein expression was also seen in these infiltrating blasts ((**F**) ×100).

**Table 1 medicina-58-01060-t001:** Laboratory data on admission to our hospital.

Complete Blood Count			Blood Chemistry			Urinalysis		
White Blood Cells	6.6 × 10^9^	/L	Total Protein	7.5	g/dL	Color	Yellow	
Blast	2.0	%	Albumin	3.5	g/dL	pH	5.5	
Metamyelocytes	1.0	%	Aspartate transaminase	34	IU/L	Specific gravity	1.043	
Segmented neutrophils	36.0	%	Alanine transaminase	15	IU/L	Protein	−	
Rod-shaped neutrophils	13.0	%	Lactate dehydrogenase	768	IU/L	Bilirubin	−	
Eosinophils	2.0	%	Total bilirubin	1.1	mg/dL	Urobilinogen	+−	
Basophils	0.0	%	Direct bilirubin	0.2	mg/dL	Occult blood	3+	
Lymphocytes	24.0	%	Creatinine	1.52	mg/dL	Glucose	−	
Monocytes	21.0	%	Blood urea nitrogen	38.2	mg/dL	Ketone	−	
Atypical lymphocytes	1.0	%	Alkaline phosphatase	199	IU/L	Nitrite	−	
Erythroblasts	6	/100	γ-Glutamyl transpeptidase	22	IU/L	White Blood Cells/HPF	30–49	/HPF
Red blood cells	1.39 × 10^12^	L	Creatine kinase	500	mg/dL	Gram-negative rods	2+	
Hemoglobin	5.4	g/dL	Sodium	136	mEq/L	**Coagulation System**		
Hematocrit	25.0	%	Potassium	4.3	mEq/L	APTT	25.0	Sec
Mean corpuscular volume	112.2	fL	Chloride	100	mEq/L	PT-INR	1.46	
Reticulocytes	29 × 10^9^	L	C-reactive protein	13.97	mg/dL	Fibrinogen	244	mg/dL
Platelets	24 × 10^9^	L	Ferritin	1665	ng/mL	D-dimer	80.04	μg/dL

APTT, activated partial thromboplastin time; HPF, high-power field; PT-INR, prothrombin time–international normalized ratio.

## Data Availability

All the data are included in the main text.

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
