# Peer review of "Renal Leukemic Infiltration Overlapping Acute Focal Bacterial Nephritis during Myelodysplastic Syndrome: An Autopsy Case Report"

_medicina, 2022, doi:10.3390/medicina58081060_

Round 1

Reviewer 1 Report

An interesting case, with myeloid sarcoma involving the kidney but no AML in peripheral blood or in bone marrow, at least initially. 

The true incidence of isolated myeloid sarcoma is not clear, as there is heterogeneity in reporting, but not all that uncommon. 

Early diagnosis after entertaining high degree of suspicion, and promptly instituting appropriate anti-neoplastic chemotherapy is vital.

Often invasive procedures, especially involving the kidney, is avoided for concerns of catastrophic bleeding complications, thus delaying the diagnosis. 

summary outlining the aim of the paper:   The authors highlight a rare presentation of extra -medullary acute myeloid leukemia, presenting as a renal lesion, in the setting of urinary tract infection/ acute focal bacterial nephritis. AML , or acute myeloid leukemia, commonly manifests as circulating blasts in the peripheral blood, and/or elevated blasts in the bone marrow (> 20%). In this case, both the blood and the marrow findings did not meet the criteria for AML.   

Main contributions   --The authors bring to notice the uncommon manifestation of acute myeloid leukemia, presenting as focal renal lesion --This is backed by the literature search, which has few mentions of this particular presentation of extramedullary AML  --They highlight that it is important to obtain tissue biopsy, in this case, renal biopsy, when clinical picture is inconsistent with the initial diagnosis of acute focal bacterial nephritis --Prompt diagnosis and treatment of acute myeloid leukemia can be life saving, as in this case the diagnosis was made on the autopsy specimen --Extra medullary AML ( myeloid sarcoma) is very challenging, and especially so if it involves difficult to biopsy / uncommonly imaged sites like the kidneys, as in this case   

Main findings    --The case brings to light an uncommon presentation of extra medullary AML involving the kidney  

Strengths   --The description, and findings, are presented in great detail, sequentially and logically --The images, and the histopathology findings of the kidney are highlighted well  

Weakness   --The English language is not upto mark in certain places, and mixes past/present tenses --Also, the autopsy finding was on day 32 or later after admission, while the initial bone marrow biopsy ( that was negative for bone marrow  involvement of AML) was on day 1. It is possible that the MDS evolved into AML in the 30 day period between the initial bone marrow biopsy and the autopsy

Author Response

Dear Reviewer 1:

Thank you for your detailed reading and suggestions on our manuscript. We are pleased that you appreciated the major instructive results of the work and recommended the publication of a revised version. Your suggestions and comments were very helpful, and during the revision phase, we have made every attempt to address your questions through clarifications in the narrative (please see point-by-point responses, below). Through addressing your suggestions, the manuscript is now much improved. With these changes, we hope that the manuscript is now acceptable to you for publication.

Weakness   --The English language is not up to mark in certain places, and mixes past/present tenses --Also, the autopsy finding was on day 32 or later after admission, while the initial bone marrow biopsy (that was negative for bone marrow involvement of AML) was on day 1. It is possible that the MDS evolved into AML in the 30-day period between the initial bone marrow biopsy and the autopsy.

Response: Thank you for highlighting the important point. English proofreading was done by MDPI’s English pre-edit service again. Furthermore, we added the sentence, “In our case, a contrast-enhanced CT image on the 15th hospitalization day revealed the remaining wedge-shaped lesion in the right kidney regardless of antibiotic administration, suggesting that renal leukemic infiltration following the transformation of MDS to AML occurred” in lines 168-171.

Reviewer 2 Report

In the present report, Murakami et al describe a case of MDS with AFBN that was unresponsive to antibiotic treatment. There are a lot of clinically relevant details included in this case report, and it is very well described. This case report illustrates the importance of clinicians reviewing the diagnosis of MDS when antibiotic treatment is ineffective in patients with AFBN. There is a novelty in this report and its significance is well described.

Author Response

Dear Reviewer 2:

Thank you for your detailed reading of our manuscript. We are pleased that you found the manuscript to be well performed, clearly presented, and informative. English proofreading was done by MDPI’s English pre-edit service again. We hope that the manuscript is now acceptable to you for publication.